# Gravimetry through non-linear optomechanics

Sofia Qvarfort [1], Alessio Serafini[1], P.F. Barker[1] & Sougato Bose[1]

Precision gravimetry is key to a number of scientific and industrial applications, including climate change research, space exploration, geological surveys and fundamental investigations into the nature of gravity. A variety of quantum systems, such as atom interferometry and on-chip-Bose–Einstein condensates have thus far been investigated to this aim. Here, we propose a new method which involves using a quantum optomechanical system for measurements of gravitational acceleration. As a proof-of-concept, we investigate the fundamental sensitivity for gravitational accelerometry of a cavity optomechanical system with a trilinear radiation pressure light-matter interaction. The phase of the optical output encodes the gravitational acceleration $g$ and is the only component which needs to be measured. We prove analytically that homodyne detection is the optimal readout method and we predict an ideal fundamental sensitivity of $\Delta g = 10^{-15}\,\mathrm{ms}^{-2}$ for state-of-the-art parameters of opto-mechanical systems, showing that they could, in principle, surpass the best atomic interferometers even for low optical intensities. Further, we show that the scheme is strikingly robust to the initial thermal state of the oscillator.

[1] Department of Physics and Astronomy, University College London, Gower Street, WC1E 6BT London, United Kingdom. Correspondence and requests for materials should be addressed to S.Q. (email: sofia.qvarfort.15@ucl.ac.uk)

  1

The practise of measuring the gravitational acceleration $g$—also known as gravimetry—has led to important advances in both fundamental science and industry. For example, local gravity variations due to mass redistribution driven by climate change have been mapped with the GRACE satellite[1–3], and more recently, the Juno spacecraft mission reported the measurement of the gravity harmonics of Jupiter[4]. Furthermore, precise measurements of $g$ can test for small deviations from Newtonian gravity on extremely small scales, which may provide indications of a deeper theory of quantum gravity[5]. In industry, precision accelerometry is extensively used in inertial navigation technologies and for conducting geological surveys.

While classical systems have long been utilised to perform accurate measurements of $g$, quantum systems offer several useful advantages, including reduced noise levels, a compact setup and most importantly an increased measurement sensitivity achieved through the power of coherence and interferometry. Over the past decade, a variety of quantum systems have been explored to this aim, in both theory and practice. The largest research effort to date has focused on atom interferometry[6–9], for which the highest achieved sensitivity currently stands at $\Delta g = 4.3 \times 10^{-9}$ ms$^{-2}$[9]. A similar investigation has been carried out for both on-chip and fountain Bose–Einstein condensate (BEC) interferometry with best sensitivity $\Delta g = 7.8 \times 10^{-10}$ ms$^{-2}$[10]. Finally, a proposal for using magnetically levitated spheres that predicts sensitivities of $2.2 \times 10^{-9}$ ms$^{-2}$ Hz$^{-1/2}$ has been put forward in[11]. For comparison, the current commercial standard is set by the LaCoste FG5-X gravimeter which can achieve a measurement sensitivity of $1.5 \times 10^{-9}$ ms$^{-2}$ Hz$^{-1/2}$[12]. More generally, the broader topic of using quantum systems to probe relativistic phenomena is currently being pursued with great interest (see for example[13–20]).

A key advantage to quantum systems are their interferometric properties. The following question arises: How can these interferometric properties be enhanced in order to improve the measurement sensitivity? One possibility is to place a quantum system in the form of a mechanical oscillator in an optical cavity, a research area known as quantum optomechanics[21]. See Fig. 1 for an illustration of a nanodiamond trapped in an optical cavity as an example of a class of optomechanical systems. The addition of the cavity allows for a strong coherent coupling between light and oscillator which, as we shall see, cancels out any initial thermal noise and fundamentally improves the measurement sensitivity of the device.

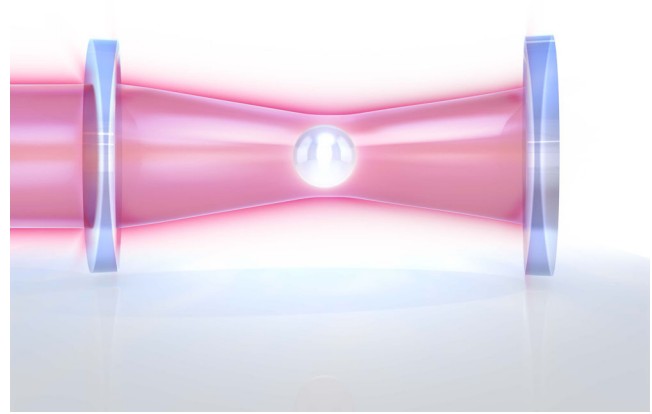

**Fig. 1** An example of an optomechanical system. A nanodiamond is trapped and laser-cooled to milliKelvin temperatures in an optical cavity

Within classical optomechanics, the idea of gravimetry and accelerometry by optically detecting the mechanical oscillator has been experimentally realised by Cervantes et al.[22]. Other avenues, such as the detection of high-frequency gravitational waves through the driving of resonant mechanical elements was proposed also in[23]. In the related field of electromechanics, Schrödinger cat states and a Kerr nonlinearity have recently been found to be useful for the same applications[24]. However, the ensuing fundamental limits on the measurement sensitivity of gravimetry in the quantum regime of optomechanics using its trilinear radiation pressure interaction is yet to be investigated. Here we undertake this task and obtain some striking results: Firstly, it is possible, in principle, to surpass the sensitivity $\Delta g$ that has been obtained in atom interferometers and other implementations to date. Secondly, due to the periodic decoupling of light and mechanics, the mechanical element does not require initial cooling to the ground state to improve the fundamental sensitivity of the gravimeter and, finally, the best possible sensitivity is achieved by a simple homodyne measurement of the cavity field, while only a low photon number in the cavity is required. That is, no measurement on the mechanical oscillator is required. Unlike the case of atomic interferometers, in optomechanics the interaction of light and matter is continuous, and we will see that our Hamiltonian cyclically entangles and disentangles the light and mechanics, leading to their decoupling. It follows that the experimental challenge will be to maintain the quantum coherence of the field and mechanics over the duration of each run of the experiment, which we set as one oscillation period of the mechanical element. This requirement, on which the plausibility of the scheme hinges, will be discussed in some detail.

## Results

**Optomechanics and Newtonian gravity.** Let us begin by considering a general optomechanical system consisting of a mechanical oscillator coupled to a light-field in the cavity. The non-gravitational Hamiltonian that describes the dynamics of an optomechanical system is given by:[25,26]

$$\hat{H} = \hbar \omega_{\rm c}\, \hat{a}^\dagger \hat{a} + \hbar \omega_m\, \hat{b}^\dagger \hat{b} - \hbar k\, \hat{a}^\dagger \hat{a}(\hat{b}^\dagger + \hat{b}), \qquad (1)$$

where $\hat{a}, \hat{a}^\dagger$ are the annihilation and creation operators for the cavity field with frequency $\omega_{\rm c}$, $\hat{b}, \hat{b}^\dagger$ are the annihilation and creation operators for the mechanical oscillator with frequency $\omega_m$, and $k$ (usually denoted $g$ in the literature, but this we shall here reserve for gravity) is a coupling constant that determines the interaction strength between the photon number $\hat{a}^\dagger \hat{a}$ and the position $\hat{x}_{\rm o} \propto (\hat{b}^\dagger + \hat{b})$ of the oscillator.

In order to introduce a coupling to a gravitational potential in the Hamiltonian, we add a term of the form $mg\,\hat{x}_{\rm o}\cos\theta$. Here, $m$ is the mass of the mechanical oscillator, $g$ is the gravitational acceleration, $\hat{x}_{\rm o} = \sqrt{\hbar/2m\omega_m}(\hat{b}^\dagger + \hat{b})$ is the position operator acting on the mechanical oscillator and $\theta$ is an angle from the vertical axis that we include in order to describe inclined systems, similar to[27]. Note that while the mass $m$ appears as a coupling in the Hamiltonian, we will later see that measurements of $g$ are mass-independent, which is what we expect from the equivalence of inertial and gravitational mass. With the addition of Newtonian gravity, the Hamiltonian of the system thus becomes

$$\hat{H}_{\rm G} = \hbar \omega_{\rm c}\, \hat{a}^\dagger \hat{a} + \hbar \omega_m\, \hat{b}^\dagger \hat{b} - \hbar k\, \hat{a}^\dagger \hat{a}(\hat{b}^\dagger + \hat{b})$$
$$+ \cos\theta\, g\sqrt{\tfrac{\hbar m}{2\omega_m}}(\hat{b}^\dagger + \hat{b}). \qquad (2)$$

**Optomechanical couplings**. While we will keep the subsequent discussion general, let us here provide three examples of common optomechanical systems and their respective coupling constants. For a a Fabry–Perot cavity with a mechanical oscillator mirror, $k$ takes the form[25,26]

$$k_{\mathrm{FP}} = \frac{\omega_{\mathrm{c}}}{L}\sqrt{\frac{\hbar}{2m\omega_m}}, \tag{3}$$

where $L$ is the length of the cavity and $m$ is the mass of the mirror. A levitated nano- or micro-crystal (e.g. a diamond or silicon bead), on the other hand, has a $k$ given by[28,29]

$$k_{\mathrm{Lev}} = \frac{P}{4V_{\mathrm{c}}\in_0}\sqrt{\frac{\hbar}{2m\omega_m}}k_{\mathrm{c}}\omega_{\mathrm{c}}, \tag{4}$$

where $\varepsilon_0$ is the permittivity of free space, $V_{\mathrm{c}}$ is the cavity mode volume, and $k_{\mathrm{c}}$ is the wavevector of the laser, given by $2\pi/\lambda$, where $\lambda$ is the laser wavelength. $P = 3V\epsilon_0(\epsilon-1)/(\epsilon+2)$ is the polarizability of the levitated object of volume $V$ and $\epsilon$ is the relative electric permittivity. Alternatively, we can also consider a BEC trapped in a cavity. Here, the collective motion of the ensemble acts as the massive oscillator. For this system, the coupling constant is given by[30,31]

$$k_{\mathrm{BEC}} = \frac{\sqrt{N}g_0^2 k_1}{\Delta_{ca}}\sqrt{\frac{\hbar}{2M\omega_m}}, \tag{5}$$

where $N$ is the number of atoms in the ensemble, $g_0$ is the single-atom cavity QED coupling rate, $M = Nm$ is the collective mass of all the trapped atoms with individual mass $m$, $k_l$ is the wavevector of the laser and $\Delta_{ca} = \omega_{\mathrm{p}} - \omega_{\mathrm{c}}$ with pumping frequency $\omega_{\mathrm{p}}$. We will return to these expressions when computing the fundamental sensitivity limits for each system in the latter part of the paper.

**System dynamics**. In order to simplify the time evolution operator $\hat{U}(t)$ corresponding to the above Hamiltonian, we rescale $\hat{H}_{\mathrm{G}}$ by dividing all terms by the oscillator frequency $\omega_m$. As a result, the time parameter $t$ now represents the labframe time multiplied by $\omega_m$, such that the oscillator has undergone a full oscillation cycle at $t = 2\pi$. The operator $\hat{U}(t)$ can then be written in the following decoupled form (see ref. [26] for details of the derivation in the absence of gravity):

$$\hat{U}(t) = \quad \exp\{-ir\hat{a}^\dagger\hat{a}t\}\exp\{i(\bar{k}\hat{a}^\dagger\hat{a} - \bar{g})^2(t - \sin t)\}$$
$$\times \exp\{(\bar{k}\hat{a}^\dagger\hat{a} - \bar{g})(\eta\hat{b}^\dagger - \eta^*\hat{b})\}\exp\{-i\hat{b}^\dagger\hat{b}\,t\}, \tag{6}$$

where $r = \omega_{\mathrm{c}}/\omega_m$, $\eta = 1 - e^{-it}$, $\bar{k} = k/\omega_m$, and $\bar{g} = \cos\theta\,g\sqrt{m/(2\hbar\omega_m^3)}$. As a rule, we will denote any dimensionless quantity with a bar. For time-dependent variables, such as dissipation rates, this means they have been rescaled with respect to $\omega_m$.

We now assume that the cavity field mode and the mechanical oscillator are initially in coherent states $|\alpha\rangle_{\mathrm{C}}$ and $|\beta\rangle_{\mathrm{O}}$ respectively. For laser light injected into the cavity, this is the natural assumption. The oscillator, on the other hand, will in reality be initialised as a thermal state, which corresponds to a random coherent state $|\beta\rangle_{\mathrm{O}}$ according to a thermal distribution. However, by starting out with a coherent state we will later argue that the gravimetric phase accumulated by the light does not depend on $|\beta\rangle_{\mathrm{O}}$ so that our procedure works equally well for an arbitrary thermal state. A formal proof of this statement can be found in

Supplementary Note 1. The initial state at $t = 0$ is then given by $|\Psi(0)\rangle = |\alpha\rangle_{\mathrm{C}}\otimes|\beta\rangle_{\mathrm{O}}$, and under $\hat{U}(t)$ it gives us the following state

$$|\Psi(t)\rangle = e^{-|\alpha|^2/2}\sum_{n=0}^{\infty}\left[\frac{\alpha^n}{\sqrt{n!}}e^{i(\bar{k}^2 n^2 - 2\bar{k}\bar{g}n)\tau}\right.$$
$$\left.\times e^{(\bar{k}n - \bar{g})(\eta\beta - \eta^*\beta^*)/2}|n\rangle_{\mathrm{C}}\otimes|\phi_n(t)\rangle_{\mathrm{O}}\right]. \tag{7}$$

where $\tau = t - \sin\,t$, and $|\phi_n(t)\rangle_{\mathrm{O}}$ are coherent states of the oscillator given by $|\phi_n(t)\rangle_{\mathrm{O}} = |e^{-it}\beta + (\bar{k}n - \bar{g})(1 - e^{-it})\rangle$. In the derivation of this state, we have adopted a rotating frame for the cavity field, thus ignoring the free evolution induced by the term $\exp\{-ira^\dagger a\}$.

The state in Eq. 7 show us that light and mechanics will entangle and disentangle periodically, with maximum entanglement occurring at $t = \pi$. At $t = 2\pi$, the oscillator state $|\phi_n\rangle_{\mathrm{O}}$ returns to $|\beta\rangle_{\mathrm{O}}$ regardless of the values of $\bar{k}, \bar{g}$ and $\beta$, and therefore by extension a thermal state also returns to its initial state because it will undergo the same compact evolution. This means that the initial oscillator state does not impact the fundamental sensitivity of this scheme. As already mentioned, a formal proof of this can be found in Supplementary Note 1. Most importantly however, at $t = 2\pi$ the cavity state is completely decoupled from the oscillator, meaning that all information about $g$ is transferred to the phase of the cavity state. As a result, any measurement scheme only needs to consider the cavity state after one oscillation period, meaning that direct or indirect access to the oscillator state is not required. This will greatly simplifies an experimental implementation, as measuring the oscillator state is generally difficult. This convenient property arises from the interferometric properties of the oscillator; its quantum nature allows it to acts as an interferometer to ensure that any initial thermal noise is removed from the cavity field, and thereby our scheme does not require cooling of the oscillator to a pure ground state. In other words, our results are valid for both coherent and thermal states. Note however that decoherence ensuing from damping to the oscillator motion during the state evolution will adversely affect the final measurement sensitivity and cause the oscillator state to grow increasingly mixed. We will not consider this kind of decoherence in this work, and instead assume that the mechanical element remains coherent over one oscillation period.

We can visualise some of the dynamics of the state in Eq. 7 by computing the expectation values of the field quadratures $\hat{X}_{\mathrm{c}} = (\hat{a}^\dagger + \hat{a})/\sqrt{2}$ and $\hat{P}_{\mathrm{c}} = i(\hat{a}^\dagger - \hat{a})/\sqrt{2}$ [32]. We focus on the cavity state, which we obtain by tracing out the oscillator. The traced-out cavity state is given by

$$\rho_{\mathrm{C}}(t) = e^{-|\alpha|^2}\sum_{n,n'}^{\infty}\left[\frac{\alpha^n(\alpha^*)^{n'}}{\sqrt{n!n'!}}e^{i(\bar{k}^2(n^2 - n'^2) - 2\bar{k}\bar{g}(n - n'))\tau}\right.$$
$$\times e^{(\bar{k}(n - n') - \bar{g})(\eta\beta - \eta^*\beta^*)/2}$$
$$\left.\times e^{-|\phi_n|^2/2 - |\phi_{n'}|^2/2 + \phi_{n'}^*\phi_n}|n\rangle\langle n'|\right]. \tag{8}$$

For decoherence-free evolution, the trajectories traced out by the system in phase space can be seen for different values of $\bar{g}$ in Fig. 2a with $\bar{k} = \bar{g} = 1$ and Fig. 2b with $\bar{k} = 1, \bar{g} = 2$. Both figures are plotted with $\alpha = \beta = 1$. We observe that the system performs increasingly complex trajectories for larger values of $\bar{g}$.

We noted above that the light and mechanics periodically entangle and disentangle during its evolution. In order to see this more clearly, we can compute the linear entropy $S(t)$ for the traced-out cavity state $\rho_c(t)$ in Eq. 8. The linear entropy is defined

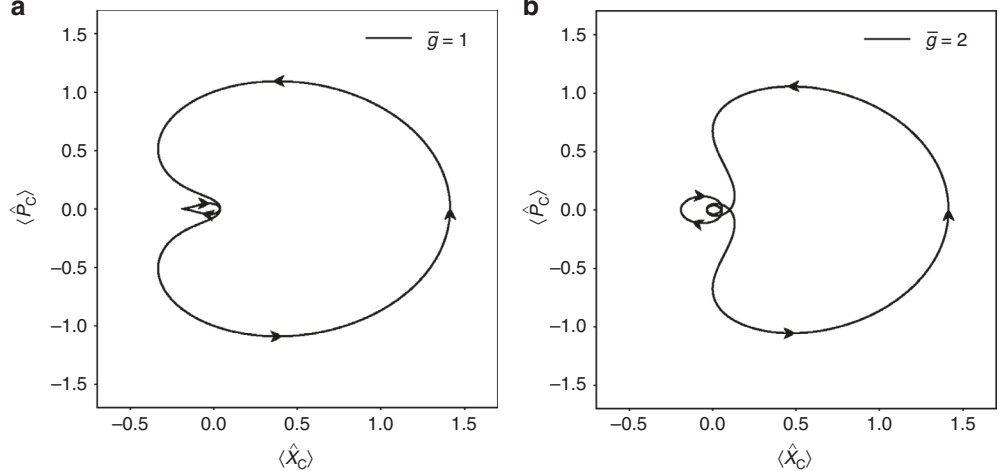

**Fig. 2** Trajectories in phase space. Figure showing the position $\langle \hat{X}_c \rangle$ and momentum $\langle \hat{P}_c \rangle$ quadratures of the cavity state $\rho_C$ that starts as a coherent state for different values of $\bar{g}$. As $\bar{g}$ grows, we see the system perform increasingly complex trajectories in phase space. Parameters used here are $\alpha = \beta = 1$, and **a** $\bar{k} = 1, \bar{g} = 1$ and **b** $\bar{k} = 1, \bar{g} = 2$

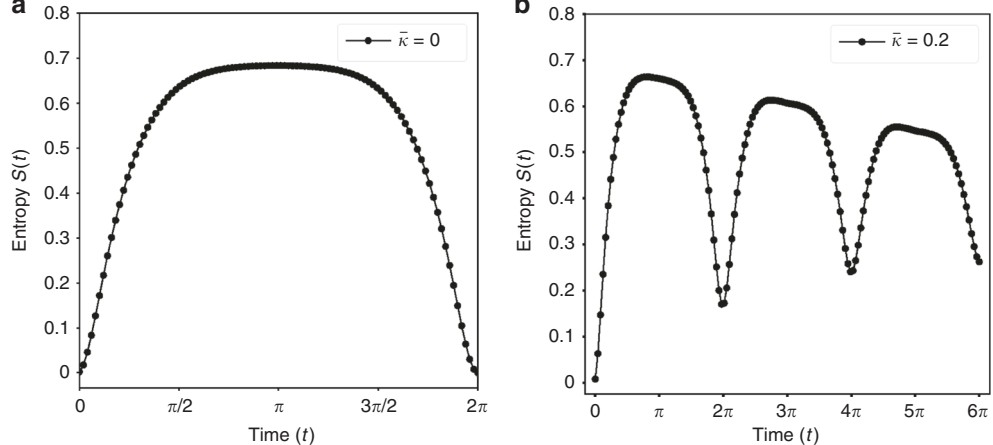

**Fig. 3** Linear entropy of the traced-out cavity state. Plots showing the linear entropy $S(t)$ for **a** free evolution and **b** noisy evolution with photon dissipation rate $\bar{\kappa} = \kappa/\omega_m = 0.2$. Both cases have been plotted for values $\bar{\kappa} = \bar{g} = \alpha = 1$

as

$$S(t) = 1 - \mathrm{tr}\left[\rho_C^2(t)\right]. \quad (9)$$

The linear entropy tells us about the entanglement between the cavity and oscillator states. The results can be found in Fig. 3a for pure state and in Fig. 3b for states undergoing decoherence with photon dissipation rate $\bar{\kappa} = \kappa/\omega_m$. We see that $S(t)$ increases until the state is maximally entangled at $t = \pi$. While a pure state completely decouples the light and mechanics at $t = 2\pi$ for any values of $\bar{k}$ and $\bar{g}$, a decohering state becomes increasingly mixed and does not return to its original state.

**Quantum metrology.** We now come to our main results which concern the use of optomechanical systems as gravimeters. The question we wish to answer is: what is the best fundamental sensitivity $\Delta g$ with which an optomechanical system can measure the gravitational acceleration $g$? Here, $\Delta g$ denotes the standard deviation of a gravimetric measurement. We can directly predict $\Delta g$ from the system's dynamics by calculating the Fisher information $I_F(t)$ which provides a natural lower bound on the variance $\mathrm{Var}(g)$ of an unknown parameter, in our case $g$. This

relationship is captured by the Cramér-Rao inequality[33–35]

$$\mathrm{Var}(g) \geq \frac{1}{N\,I_F(t)}, \quad (10)$$

where $N$ is the number of measurements. Thus if we maximise $I_F(t)$, we minimise the measurement spread of $g$.

**Quantum Fisher information.** The Fisher information comes in two forms: the measurement-specific classical Fisher information (CFI) and the quantum Fisher information (QFI). The QFI, which we denote $H_Q(t)$, is computed by optimising over all possible positive-operator valued measures (POVMs) and their resulting CFI[36]. Thus $H_Q$ represents the ultimate bound on obtainable information from a system, but it does not reveal which specific measurement is required to achieve it. For a general mixed quantum state $\rho(t)$ the QFI is given by $H_Q(t) = \left[\rho(t)\mathcal{L}^2\right]$, where $\mathcal{L}$ is the symmetric logarithmic derivative. In general, it is difficult to obtain $\mathcal{L}$ analytically, especially for noisy systems. There are however methods for finding a noisy bound on the Cramér-Rao inequality[37]. A similar method for many-body systems was proposed in[38], and numerical methods were shown to be effective for a class of specific systems[39]. We shall not be using these methods here, as we shall instead investigate specific measurements for the noisy scenario to better

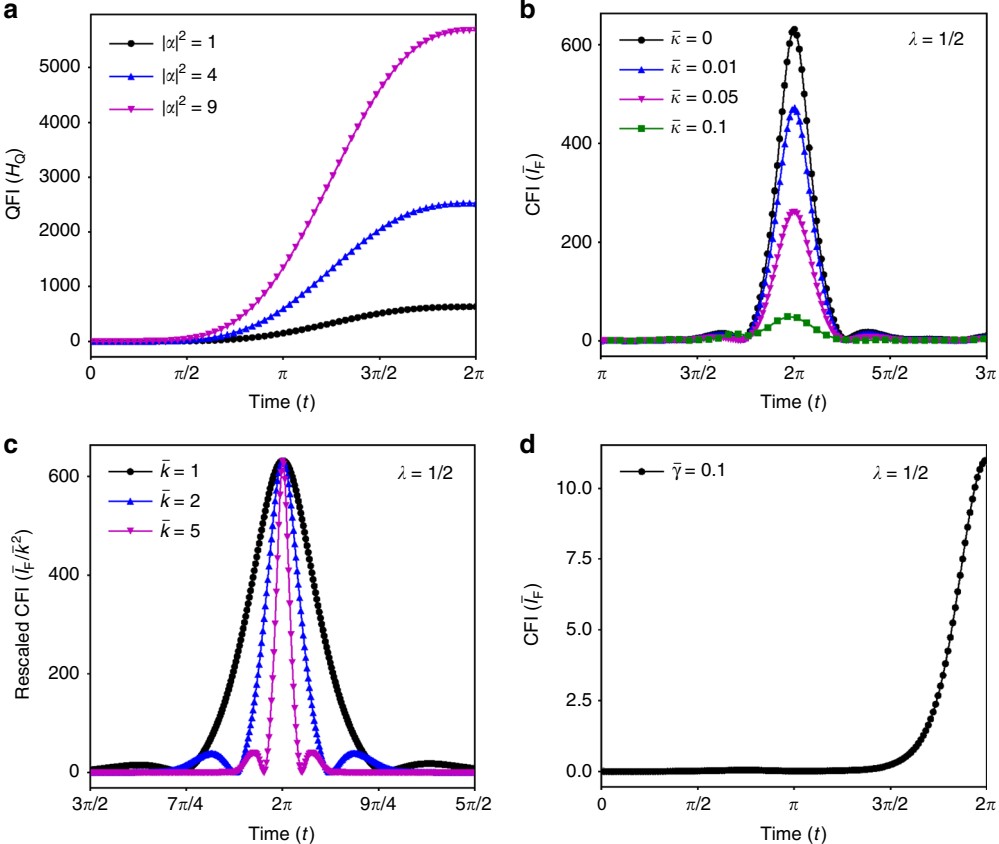

**Fig. 4** Fisher information for measurements of gravitational acceleration $g$. Plots of the quantum Fisher information (QFI) and classical Fisher information (CFI) for measurements of $g$. **a** shows the QFI $H_Q(t)$ for 1, 4 and 9 photons $|\alpha|^2$ with rescaled couplings $\bar{k} = \bar{g} = 1$. **b** shows the dimensionless CFI $\bar{I}_F(t)$ with and without photon dissipation rate $\bar{\kappa} = \kappa/\omega_m$ for a momentum measurement ($\lambda = 1/2$) and $\bar{k} = \bar{g} = 1$. **c** shows how the peak of the CFI for a momentum measurement narrows with increasing $\bar{k}$ and with constant $\bar{g} = 1$. **d** shows the $I_F(t)$ for a momentum measurement of photons that leak from the cavity with environmental coupling $\bar{\gamma} = 0.1$ and $\bar{k} = \bar{g} = 1$. All plots use parameter $\beta = 1$

approximate an experimental setting. This will later allow us to prove the optimality of the homodyne measurement.

Let us start by deriving a fundamental bound to the sensitivity. We specialise to the simpler case where the state $\rho(t)$ is pure. Setting $\rho(t) = |\Psi(t)\rangle\langle\Psi(t)|$ the QFI becomes

$$H_Q(t) = 4\left[\langle\partial_g\Psi(t)|\partial_g\Psi(t)\rangle - \left|\langle\Psi(t)|\partial_g\Psi(t)\rangle\right|^2\right], \quad (11)$$

where we have used the notation $\partial_g = \partial/\partial_g$.

At first glance, the QFI of the global system might not seem very relevant as the mechanical part of the optomechanical system cannot easily be measured directly. However, we recall that the coherent state $|\phi_n(t)\rangle_O$ returns to $|\beta\rangle_O$ at $t = 2\pi$, so that all information about $g$ is transferred to the phase of the pure, decoupled cavity state. Since the decoupling time does not depend on $\beta$, this is also the case for a thermal state that may be written as a statistical mixture of coherent states (see Supplementary Note 1 for a proof of this statement). Calculating the QFI for this state will therefore provide an experimentally accessible notion of the fundamental sensitivity of the device. We find the following expression for $H_Q(t)$ at $t = 2\pi$:

$$H_Q(2\pi) = \frac{32\pi^2\bar{k}^2 m|\alpha|^2 \cos^2\theta}{\hbar\omega_m^3}. \quad (12)$$

Note that the mass term $m$ is cancelled by the appearance of $\sqrt{m}$ in the coupling constant $\bar{k}$, so that the final accelerometry

measurement will be mass-independent. We also note the strong dependence on $\bar{k}$ and $\omega_m$, and that the expression scales linearly with the number of photons $|\alpha|^2$.

To find $H_Q(t)$ for the global state at any time $t$ we resort to numerical calculations. We consider the case $\bar{k} = \bar{g} = 1$ to allow for future comparisons with subsequent numerical evaluation of the CFI, which will be restricted to the same narrow parameter range. The resulting $H_Q(t)$ as a function of $t$ can be found in Fig. 4a with $\bar{k} = \bar{g} = 1$ and $\beta = 1$ for various values of $|\alpha|^2$. We note that $H_Q(t)$ reaches its maximum value at $t = 2\pi$, which means that Eq. 12 returns the largest possible value during one oscillation period for any choice of system.

**Classical Fisher information.** Let us now consider a specific measurement of $g$. The CFI $I_F(t)$ determines the minimum standard deviation of a parameter estimator once we have chosen a single specific measurement with POVM elements $\{\Pi_x\}$. The CFI is given by the expression

$$I_F(t) = \int dx \frac{1}{p(x|g)}\left(\frac{\partial p(x|g)}{\partial g}\right)^2, \quad (13)$$

where $p(x|g) = \text{tr}[\Pi_x\rho(g)]$ is a conditional probability distribution.

We now consider a general homodyne measurement on the traced-out cavity state $\rho_C$. For notational convenience, we use a general Hermitian operator $\hat{x}_\lambda = (\hat{a}\exp\{-i\lambda\} + \hat{a}^\dagger\exp\{i\lambda\})/\sqrt{2}$, where $\lambda$ denotes a label that rotates between the field quadratures[40]. Any two operators that differ by $\lambda = \pi/2$ form a

conjugate pair which satisfies the position-momentum commutator relation. In the following, we shall refer to the choices $\lambda = 0$ and $\lambda = \pi/2$ as a position and momentum measurement respectively. In order to calculate $I_F(t)$ we must find the probability distribution $p(x_\lambda | g) = \mathrm{tr}[|x_\lambda\rangle\langle x_\lambda|\rho_c(g)]$, where $|x_\lambda\rangle\langle x_\lambda|$ are the eigenstate of $\hat{x}_\lambda$. While the position eigenstates themselves are not proper vectors, we can make use of a standard result from the quantum harmonic oscillator: $\langle n|x_\lambda\rangle = \pi^{-1/4}2^{-n/2}(n!)^{-1/2}\exp\{-x_\lambda^2/2\}H_n(x_\lambda)\exp\{in\lambda\}$[40], to write

$$
\begin{aligned}
p(x_\lambda|g) = \quad & e^{-|\alpha|^2}\sum_{n,n'}\Bigg[\frac{\alpha^n(\alpha^*)^{n'}}{\sqrt{n!n'!}}e^{i(\bar{k}^2(n^2-n'^2)-2\bar{k}\bar{g}(n-n'))\tau} \\
& \times \frac{e^{-x_\lambda^2}}{\pi^{1/2}}\frac{H_n(x_\lambda)H_{n'}(x_\lambda)e^{-i\lambda(n-n')}}{2^{(n+n')/2}\sqrt{n!n'!}} \\
& \times e^{(\bar{k}(n-n')-\bar{g})(\eta\beta-\eta^*\beta^*)/2} \\
& \times e^{-|\phi_n|^2/2-|\phi_{n'}|^2/2+\phi_{n'}^*\phi_n}\Bigg],
\end{aligned}
\tag{14}
$$

where $H_n(x)$ are the Hermite polynomials of order $n$. These probabilities in turn gives rise to a CFI of the form

$$
\begin{aligned}
I_F(t) = \quad & \cos^2\theta\frac{m}{2\hbar\omega_m^3}\left(-4\bar{k}^2\tau^2\right)e^{-|\alpha|^2} \\
& \times \int dx_\lambda \frac{\left[\sum_{n,n'}(n-n')c_{n,n'}d_{n,n'}(x_\lambda)f_{n,n'}\right]^2}{\sum_{n,n'}c_{n,n'}d_{n,n'}(x_\lambda)f_{n,n'}},
\end{aligned}
\tag{15}
$$

where

$$
c_{n,n'} = \frac{(\alpha^*)^{n'}\alpha^n}{\sqrt{n!n'!}}e^{i(\bar{k}^2(n^2-n'^2)-2\bar{k}\bar{g}(n-n'))\tau},
\tag{16A}
$$

$$
d_{n,n'}(x_\lambda) = \frac{e^{-x_\lambda^2}}{\pi^{1/2}}\frac{H_n(x_\lambda)H_{n'}(x_\lambda)e^{-i\lambda(n-n')}}{2^{(n+n')/2}\sqrt{n!n'!}},
\tag{16B}
$$

$$
f_{n,n'} = e^{(\bar{k}(n-n')-\bar{g})(\eta\beta-\eta^*\beta^*)/2}\times e^{-|\phi_{n'}|^2/2-|\phi_n|^2/2+\phi_{n'}^*\phi_n}.
\tag{16C}
$$

**Timescales of measurements**. Let us analyse the expression for $I_F(t)$. We immediately note that any terms in the sum with $n = n'$ do not contribute to the Fisher information. The remaining behaviour of $I_F$ can be inferred from the second exponential in $f_{n,n'}$, namely $\exp\{-|\phi_{n'}|^2/2 - |\phi_n|^2/2 + \phi_{n'}^*\phi_n\}$ as this will dominate the entire expression for large $\bar{k}$. If we simplify the expression in the exponential, we find that it is equal to

$$
\exp\left\{-\bar{k}^2(n-n')^2(1-\cos t) + \frac{\bar{k}(n-n')}{2}[\beta\eta-\beta^*\eta^*]\right\}.
\tag{17}
$$

For $n \neq n'$ and large $\bar{k}$, the first term will dominate, and the exponential will be small for any $t$ that is not a multiple of $2\pi$. In other words, the Fisher information for a homodyne measurement becomes significant only when light and mechanics are completely decoupled. Figure 4c shows how the CFI for a momentum measurement (with $\lambda = \pi/2$) for $\bar{g} = \alpha = \beta = 1$ and $\bar{k} = 1, 2, 5$ becomes increasingly narrow as $\bar{k}$ grows larger. For clarity, we have rescaled $I_F$ with $\bar{k}$ in the plot. Note that for small $\bar{k}$ we still find large $I_F$ at times $t \neq 2\pi$.

We saw earlier that the QFI scales with $\bar{k}^2$, which mean that the scheme favours systems with a large single-photon coupling. We shall soon show that the CFI coincides with the QFI at $t = 2\pi$, but in the meantime we must explore what the narrowing of the CFI at $t = 2\pi$ entails for our measurement scheme. The narrow peak of the CFI will require the homodyne measurement to be performed within an increasingly narrow time-window. We can estimate the timescale in question by finding the full-width-half-maximum (FWHM) of the peak. To do so, we consider only the dominant first term $-\bar{k}^2(n-n')^2(1-\cos t)$ for small perturbations in $t$ around $t = 2\pi$, thus $\cos(2\pi + t') \approx 1 - t'^2/2$. That brings the first term into the form $-\bar{k}^2(n-n')^2 t'^2/2$, which is now a Gaussian distribution. For a Gaussian function with $\exp\{-(x - x_0)^2/(2\sigma^2)\}$, the FWHM is given by $2\sqrt{2\ln 2}\,\sigma$. In our case, we find $\sigma^2 = (2\bar{k}^2(n-n')^2)^{-1}$. We already noted that terms with $(n - n')$ will not contribute to the CFI, and any term with $|n - n' \gg 1|$ will just cause the peak to narrow further. Thus we only consider the terms with $|n - n'| = 1$, leaving us with $\sigma = (2\bar{k})^{-1}$, and so we conclude that any measurement must be performed roughly on a timescale of $(\omega_m\bar{k})^{-1} = k^{-1}$.

**Optimality of homodyne detection**. Let us see if we can simplify the expression for $I_F(t)$ even further and whether it bears any semblance to the QFI. At $t = 2\pi$, $\phi_n(2\pi) = \beta$ and $\eta = 0$. Then setting $\bar{k}$ and $\bar{g}$ to integer values causes $I_F(2\pi)$ to lose all dependence of $\bar{g}$. The coefficients reduce to $c_{n,n'} = (\alpha^*)^{n'}\alpha^n/\sqrt{n!n'!}$ and $f_{n,n'} = 1$. We now consider the generating function for the Hermite polynomials $e^{2xt-t^2} = \sum_{n=0}^{\infty}t^n H_n/n!$. Taking the derivative of both sides results in $(2x - 2t)e^{2xt-t^2} = \sum_{n=n'}^{\infty}t^{n-n'}H_n/(n - n')!$, which we can use to show that Eq. 15 reduces to the compact expression

$$
I_F(2\pi) = \frac{8\pi^2\bar{k}^2 m}{\hbar\omega_m^3}\left(ie^{-i\lambda}\alpha - ie^{i\lambda}\alpha^*\right)^2.
\tag{18}
$$

This expression coincides precisely with the QFI in Eq. 12 for complementary choices of $\lambda$ and $\alpha$. To better see why, we rewrite the term in the brackets as $\left[(e^{-i\lambda} - e^{i\lambda})i\,\mathrm{Re}\{\alpha\} - (e^{-i\lambda} + e^{i\lambda})\mathrm{Im}\{\alpha\}\right]^2$. We now note that when $\lambda = 0$, only $\mathrm{Im}\{\alpha\}$ contributes to the CFI, whereas at $\lambda = \pi/2$, only $\mathrm{Re}\{\alpha\}$ contributes. For both of these specific choices of $\lambda$, and when matched by $\alpha$ being either entirely real or imaginary, the CFI coincides precisely with the QFI in Eq. 12 because the term in the brackets reduces to $4\mathrm{Re}\{\alpha\}^2$ or $4\mathrm{Im}\{\alpha\}^2$, respectively. We conclude that the homodyne measurement saturates the QFI limit up to a phase dependence of $\alpha$, which can always be accounted for by changing the quadrature of the homodyne measurement. Note, however, that at other times than $t = 2\pi$. the homodyne measurement will be zero for all choices of $\lambda$ and $\alpha$, and so it only saturates the QFI when the light and mechanics have decoupled.

Finally, the absence of $\bar{g}$ from $I_F(2\pi)$ is not a problem for sensing $g$—it just means that the sensitivity at times $t = 2\pi$ is independent of the actual value of $g$. Numerical analysis suggests that larger values for $\bar{g}$ causes the CFI to oscillate increasingly quickly before reaching its maximum value (see Supplementary Note 2). The optimality of the homodyne detection for sensing within our scheme is greatly advantageous as it is a routine measurement which is easy to accomplish. It has in fact also been shown to be an optimal measurement[41] in other contexts.

**Decoherence**. The calculation above is valid for pure states, but in practice every measurement will suffer various forms of decoherence. We will here investigate the effects of decoherence on the CFI for a narrow parameter range, as realistic parameters are very

difficult to simulate numerically. We shall later use these results as indications of the behaviour of realistic systems.

There exists a large variety of decoherence effects for optomechanical systems, such as decoherence due to photons leaking from the cavity, or phonons gradually being lost from the mechanical element. The latter manifests as a gradual damping of the oscillator motion, which moves the state towards a mixture in the coherent state basis[42–45]. This problem has previously been treated analytically, which is possible because the decoherence operators commute with the Hamiltonian. Thus we refer to these works and will not treat the mechanical decoherence here. Instead, we make the assumption that the phonon decoherence is negligible over one oscillation period of the oscillator.

The effect of photons leaking from a cavity on a state $\rho(t)$ can be modelled using a Lindblad master equation of the form

$$\frac{\partial \rho(t)}{\partial t} = -\frac{i}{\hbar}\left[\hat{H}, \rho(t)\right] + \hat{L}\rho(t)\hat{L}^\dagger - \frac{1}{2}\left\{\rho(t), \hat{L}^\dagger\hat{L}\right\}, \quad (19)$$

where $\{\cdot, \cdot\}$ denotes the anti-commutator, $\hat{L} = \sqrt{\bar{\kappa}}\,\hat{a}$ are Lindblad operators and $\bar{\kappa} = \kappa/\omega_m$ is the decoherence rate $\kappa$ with respect to the rescaled time $t$. This equation cannot easily be solved analytically since the operator $\hat{a}$ does not commute with the Hamiltonian $\hat{H}_G$ in Eq. 2. Some solutions have been found for specific cases, for example when assuming that the photon leakage occurs only during the injection of the state into the cavity. The decoherence can then be modelled as a series of beamsplitters[46]. We will not consider these modifications here, but instead solve the Lindblad master equation numerically and compute the Fisher information $I_F(t)$ for the resulting mixed state.

In all subsequent numerical evaluations, we will set $\bar{k} = \bar{g} = 1$ and $\alpha = 1$ (note the choice of $\alpha \in \mathbb{R}$, which will optimise the CFI for $\lambda = \pi/2$). Larger values will cause the system to quickly grow numerically unstable due to the inclusion of non-linear terms such as $(a^\dagger a)^2$ in the evolution in Eq. 6. While $\bar{k} = 1$ is experimentally achievable with the right choice of parameters, we can justify setting $\bar{g} = 1$ by noting that it physically corresponds to a heavily inclined cavity with $\theta \approx \pi/2$. Since we are interested in the general behaviour of the CFI under decoherence, we will here be working with the dimensionless Fisher information $\bar{I}_F(t)$ (see the Methods section). Thus these numerical investigations should only be seen as a indication as to how decoherence will affect $I_F(t)$, and not as predictions for the sensitivity of a realised device. We shall later extrapolate from these results to make a prediction for realistic systems.

The behaviour of $\bar{I}_F(t)$ can be found in Fig. 4b for a momentum measurements with $\lambda = \pi/2$. A measurement with $\lambda = 0$ and $\alpha = i$ would show the same results. We note that larger values of $\bar{\kappa}$ do affect the CFI adversely, but setting $\bar{\kappa} = 0.1$ implies that about 10% of the pure-state CFI is still accessible.

**Measurements of leaking photons.** In practise, a homodyne measurement is performed by monitoring and measuring the photons that continuously leak from the cavity. Aside from the experimental considerations, such a scheme also negates part of the photon dissipation considered above. We briefly estimate the CFI obtained through such a setup by using a simplified model where a pure vacuum state of the environment $|0\rangle_E$ is added to our original state $|\Psi(t)\rangle_{CO}$, giving us the combined initial state $|\Psi(t)\rangle_{CO} \otimes |0\rangle_E$. We then add a rotating wave interaction term $\hat{H}_I$ to the Hamiltonian $\hat{H}_G$ in Eq. 2, of the form

$$\hat{H}_I = \gamma(\hat{a}^\dagger\hat{c} + \hat{a}\hat{c}^\dagger), \quad (20)$$

where $\gamma$ is the interaction strength and $\hat{c}$ and $\hat{c}^\dagger$ are the creation and annihilation operators of the environment. The effect of this

interaction Hamiltonian is to couple the cavity state to the environment which causes information about $g$ to slowly leak out from the cavity into $|0\rangle_E$.

As before, we evolve the full state for a single-photon $|\alpha|^2 = 1$ and with parameters $\bar{k} = \bar{g} = 1$. To maximise the CFI, we choose $\alpha \in \mathbb{R}$ and $\lambda = \pi/2$. The results can be found in Fig. 4d for a rescaled coupling strength $\bar{\gamma} = 0.1$, where $\bar{\gamma} = \gamma/\omega_m$. As evident from Fig. 4d, we suffer a $10^{-2}$ reduction in the information that can be extracted from the system. Note also that the behaviour of $I_F(t)$ for this scenario will most likely also resemble a delta function centred around $t = 2\pi$ for realistic parameters. Additional plots for this simplified model can be found in Supplementary Note 3.

**Ideal sensitivities.** In this section we shall first calculate the ideal Fisher information for the three optomechanical systems considered above, and then discuss the experimental challenges and advantages to an optomechanical gravimeter. As we here calculate the fundamental sensitivity, which is unlikely to be realised, we will only concern ourselves with order-of-magnitude estimates. These results are meant to showcase the potential of optomechanical systems, and to do so we have chosen state-of-the-art parameters that have been implemented in a variety of systems. For discussions of an experimental implementation including noise, see the Discussion.

Starting with the Fabry–Perot cavity system, we choose a fully vertical cavity with $\theta = 0$ and use the following state-of-the-art experimental parameters: We choose a mass $m = 10^{-6}$ kg, oscillator frequency $\omega_m = 10^3$ Hz, cavity frequency $\omega_c = 10^{14}$ Hz, cavity length $L = 10^{-5}$ m and a photon number of $|\alpha|^2 = 10^6$. For these values, the rescaled coupling constant in Eq. 3 becomes $\bar{k}_{FP} \approx 2.30$, which gives us a Fisher information of $I_F = 1.58 \times 10^{28}$ m$^{-2}$ s$^4$. This implies a sensitivity of $\Delta g \approx 7.96 \times 10^{-15}$ ms$^{-2}$.

Next, we look at a levitated micro-object confined in an ion trap interacting with an optical cavity, as demonstrated very recently in refs[47,48]. Again setting $\theta = 0$ for maximal effect, we use mass $m = 10^{-14}$ kg, oscillator frequency $\omega_m = 10^2$ Hz, cavity frequency $\omega_c = 10^{14}$ Hz, volume $V = 10^{-18}$ m$^3$, cavity mode volume $V_c = 10^{-14}$ m$^3$, electric permittivity $\varepsilon = 5.7$ for nanodiamonds, laser wavelength $\lambda = 1064 \times 10^{-9}$ m and a photon number of $|\alpha|^2 = 10^6$. From these values we obtain $\bar{k}_{Lev} = 1963$, which leads to $I_F = 1.15 \times 10^{29}$ m$^{-2}$s$^4$. This gives us a final sensitivity of $\Delta g \approx 2.94 \times 10^{-15}$ ms$^{-2}$ for levitated nanospheres.

Finally, let us also consider cold atoms trapped in a cavity. Based on[30], we choose the following parameters: a wavelength $\lambda = 780$ nm, implying $\omega_c = 10^{15}$ Hz, a single-atom coupling of $g_0 = 10^7$ Hz, an atomic oscillation frequency $\omega_m = 10^2$ Hz, a single-atom mass $m = 10^{-25}$ kg, a detuning of $\Delta_{ca} = 10^{11}$ Hz, and a laser wavevector of $k_l = 10^8$ m$^{-1}$. With $N = 10^5$ atoms trapped in the cavity, we find that $\bar{k}_{BEC} = 2.30 \times 10^6$ and $I_F = 1.58 \times 10^{19}$ m$^{-2}$ s$^4$, giving a sensitivity of $\Delta g \approx 2.5 \times 10^{-10}$ ms$^{-2}$. The reason for this disparity seems to be that the polarisability of the collection of cold atoms is not high enough to match the polarisability exhibited by the nanosphere. The number of trapped atoms can hardly match the number of atoms in a single nanosphere. One would either have to increase the number of atoms trapped in the cavity or increase the single-atom coupling strength to increase the Fisher information.

**Comparison of theoretical results.** Let us briefly compare the results obtained here with the performance of other quantum systems in the literature. In Table 1 we have listed a variety of experimentally implemented gravimeter systems with their best achieved sensitivity to date. Table 2, on the other hand, lists the

**Table 1 Comparison between gravimetry sensitivities obtained by various experimental systems**

**Experiments**

| System | $\Delta g$ | $\Delta g/\sqrt{Hz}$ | Int. time |
|---|---|---|---|
| LaCoste FG5-X[12] | $1 \times 10^{-9}$ | $1.5 \times 10^{-7}$ | 6.25 h |
| Atom intf[9]. | $5 \times 10^{-9}$ | $4.2 \times 10^{-8}$ | 100 s |
| On-chip BEC[10] | $7.8 \times 10^{-10}$ | $5.3 \times 10^{-9}$ | 100 s |
| Optomech. accel[22]. | $3.10 \times 10^{-5}$ | $9.81 \times 10^{-7}$ | $10^{-3}$ s[a] |

These include the commercial LaCoste FG5-X, atom interferometry, gravimetry through on-chip Bose-Einstein condensate (BEC) and classical optomechanical accelerometry. The second column lists the sensitivity $\Delta g$ in ms$^{-2}$ and the third column lists the $\sqrt{Hz}$-noise $\Delta g/\sqrt{Hz}$ in ms$^{-2}$Hz$^{-1/2}$. The last column indicates the integration time needed to achieve each sensitivity
[a]This value was provided to us by the authors of ref.[22]

**Table 2 Comparison between sensitivities obtained by theoretical predictions for a variety of systems**

**Theoretical predictions**

| System | $\Delta g$ | $\Delta g/\sqrt{Hz}$ | Cycle time |
|---|---|---|---|
| Magnetomech[11] | $2.2 \times 10^{-7}$ | $2.2 \times 10^{-9}$ | $10^{-4}$ s |
| Fabry–Perot optomech[a] | $10^{-15}$ | $10^{-16}$ | $10^{-3}$ s |
| Levitated optomech[a] | $10^{-15}$ | $10^{-16}$ | $10^{-2}$ s |
| Cold atoms[a] | $10^{-10}$ | $10^{-11}$ | $10^{-2}$ s |

These include magnetomechanics, a Fabry–Perot optomechanical system, a levitated nanosphere optomechanical system and trapped cold atoms. The second column lists the sensitivity $\Delta g$ in ms$^{-2}$ and the third column lists the $\sqrt{Hz}$-noise $\Delta g/\sqrt{Hz}$ in ms$^{-2}$Hz$^{-1/2}$. The last column indicates the cycle time or oscillation frequency $\omega_m$ for each system
[a]Values calculated in this work are denoted

ideal fundamental limits to sensitivities calculated in this work and others. The values for $\Delta g$ and $\Delta g/\sqrt{Hz}$ are presented in units of ms$^{-2}$ and ms$^{-2}$ Hz$^{-1/2}$, respectively. The last column in Table 1 lists the integration time for each experiment, whereas in Table 2 the last column lists the experimental cycle time set by the oscillation frequency of the system in question. For atom interferometry, it is suggested in[49] that sensitivities of $\Delta g \sim 10^{-12}$ ms$^{-2}$ might be achieved, and a study of the fundamental limits has very recently been presented in[50].

## Discussion

In this work, we investigated a new scheme for measurements of the gravitational acceleration $g$ using a compact cavity optomechanical system with the usual trilinear optomechanical coupling to the cavity field. We derived a fundamental limit to the sensitivity $\Delta g$ by computing the QFI and showed that the optimal sensitivity is achieved by a homodyne detection scheme performed on the cavity state at time $t = 2\pi$. That is, no direct measurement of the mechanical oscillator is required. Using the expression in Eq. 13 and state-of-the-art experimental parameters, we predict a upper bound on the sensitivity of order $\Delta g \sim 10^{-15}$ ms$^{-2}$ for both a Fabry–Perot cavity and a levitated microsphere cavity, and $\Delta g \sim 10^{-10}$ ms$^{-2}$ for trapped cold atoms. These values compare favourably to all other currently available experimental and theoretical gravimetry proposals (see Tables 1 and 2). Furthermore, the quantum nature of the oscillator ensures that any thermal distribution in its initial state does not affect the fundamental sensitivity. However, as our scheme relies on superpositions involving distinct coherent states, we require thermal decoherence during one period of the oscillator motion to be negligible, which we estimate requires a $Q$-factor of at least $10^6$ for the case of a Fabry–Perot cavity (see below). To explore the effects of photons leaking from the cavity, we numerically

explored a narrow parameter range with $\bar{k} = \bar{g} = 1$, which physically corresponds to a nearly horizontally aligned cavity. We found that this form of decoherence does affect the system's performance, but not severely. Finally, we briefly investigated what proportion of $\Delta g$ we retain by performing measurements on the photons that leak from the cavity. Using a simplified noise model, we found a reduction of $10^{-2}$ in the resulting Fisher information. Given these results, we believe that there is significant potential in the use of quantum optomechanical systems for measurements of gravity and acceleration.

Let us now address some of the experimental challenges related to this scheme. Due to measurement inefficiencies and additional sources of decoherence not considered here, the final performance of optomechanical systems will naturally be expected to be lower than the values presented in Table 2. While we have shown that the initial optomechanical state does not need cooling to the ground state, thermal noise due to external influences during the evolution will gradually decohere the oscillator motion. We estimate that in the case of a Fabry–Perot cavity cooled to a temperature of milliKelvin, a number of $\hbar\omega_m/(k_BT_{th}) = N$ phonons are present in the system at any time. Here, $k_B$ is Boltzmann's constant and $T_{th}$ is the system's temperature. To retain coherence throughout the evolution, we require that $\kappa_m N \ll \omega_m$, where $\kappa_m$ is the phonon dissipation rate. In other words, the timescale of phonon decoherence $\kappa_m$ must be much less than the characteristic timescale of the system. With $\omega_m = 1$ kHz, as we assumed for Fabry–Perot cavities, we find $N = 10^5$ and $\kappa_m = 10^{-2}$ Hz. A cavity which achieves such a decoherence rate must have a mechanical $Q$-factor of at least $Q = \omega_m/\kappa_m \sim 10^6$ to retain coherence, a regime which is not unprecedented.

Next, let us discuss which parameters yield the best sensitivities. Firstly, we note that the QFI in Eq. 12 ultimately scales with $\omega_m^{-6}$. In addition to the factor $\omega_m^{-3}$ in the denominator, we acquire an extra $\omega_m^{-2}$ from the rescaled coupling constant $\bar{k} = k/\omega_m$. The final factor of $\omega_m$ comes from the dependence of $\omega_m$ in $k^2$. Given this scaling, we require $\omega_m$ to be as small as possible. At the same time, we also require the photon dissipation rate $\kappa$ to be low. From our simulations, we saw that we require at least $\bar{\kappa} = \kappa/\omega_m = 0.1$. This combination is difficult to achieve as low $\omega_m$ means the cavity must remain coherent over longer timescales. Therefore, the main experimental challenge of this scheme is to reduce $\omega_m$ and $\kappa$ at the same time. Taking our numerical results as guidance, we essentially require that $\bar{\kappa} = \kappa/\omega_m \ll 1$, which is nothing but the resolved side-band regime[28].

In the above, we used state-of-the-art parameters to calculate the ideal QFI for a variety of systems. However, as we just saw, the photon dissipation rate $\kappa$ must be very low for these sensitivities to be achieved, and this has not yet been experimentally demonstrated for the parameters we used in Table 2. As technology improves we expect that this to be possible in future experiments, but for now, let us estimate the sensitivities that could be achieved today already. One of the best coherence times to date was demonstrated in[51], which achieved a cavity linewidth of $\kappa = 660$ Hz. To achieve a rescaled photon rate of $\bar{\kappa} = 0.1$ for this system, we let $\omega_m = 6600$ Hz and use $L = 9.4$ cm as reported in the paper. We keep $m = 10^{-6}$ kg (since the QFI is ultimately independent of mass) and let $\omega_c = 10^{14}$ Hz as before. Because the oscillation frequency $\omega_m$ is rather high, we choose to calculate $I_F$ for the Fabry–Perot cavity with a mechanical mirror, as this system performed slightly better for higher $\omega_m$. The resulting coupling constant is $\bar{k}_{FP} = 1.44 \times 10^{-5}$, and the Fisher information is $I_F \approx 2.16 \times 10^{15}$ m$^{-2}$ s$^4$. This leads to $\Delta g \approx 2.15 \times 10^{-8}$ ms$^{-2}$. If we now assume that decoherence causes a similar proportion of the Fisher information to dissipate at these parameters compared to the ones chosen in our numerical simulations, we see that we retain about 10% of the pure-state Fisher information. Using this

assumption, we find $\Delta g \approx 6.80 \times 10^{-8}$ ms$^{-2}$ and a $\sqrt{\text{Hz}}$-noise of $8.37 \times 10^{-10}$ ms$^{-2}/\sqrt{\text{Hz}}$. This is directly comparable with the values in Table 1, and so we believe that this scheme could be experimentally realised today, although the experimental challenges are of course substantial.

Let us briefly discuss ways in which we can decrease $\kappa$ further and how this might affect the Fisher information. A heuristic estimate for $\kappa$ can be given by considering the number of times per second that a single photon traverses the cavity. Each time the photon is reflected at the mirror, it has a $T = 1 - R$ chance of being transmitted instead of reflected. Here, $T$ is the proportion of transmissions and $R$ is the proportion of the number of reflections. The photon bounces off a mirror $c/L$ times per second, where $c$ is the speed of light. Thus we can take the dissipation rate to be $\kappa = Tc/L$, which means that increasing $L$ decrease the photon dissipation rate $\kappa$, as the photon is effectively spending longer inside the cavity. However, increasing $L$ also decreases the single-photon coupling constant, as we saw in the calculation above. This is true for all couplings we quote here, but it is perhaps most clearly seen for the case of the mechanical mirror and a Fabry–Perot cavity, with $k_{\text{FP}}$ given by Eq. 3. $k_{\text{FP}}$ scales with $L^{-1}$, and so do the other couplings, through their dependence on the cavity volume $V_c$ or the single-photon coupling $g_0$. We recall that the Fisher information depends on $\bar{k}^2$, which means that it ultimately scales with $L^{-2}$. Thus, changing $L$ by an order of 10 will decrease the Fisher information by an order of $10^2$. This contributes to the challenges of realising this scheme. However, it is important to note that there are realistic ways of increasing $L$ without changing the single-photon coupling: One such method was explored in[52], where $L$ was increased by adding an optical fibre to the cavity.

Furthermore, in the above we proved the optimality of a homodyne detection scheme, but we also found that such a measurement must be performed within a rather narrow temporal window, of timescale $1/k$. Let us here estimate how quickly these measurements have to be performed based on the values we calculated for the coupling constant $k$. The nanospheres displayed the highest single-photon coupling with $\bar{k}_{\text{Lev}} \times \omega_m = 10^5$ Hz for the choice of $\omega_m = 10^2$ Hz. Thus any homodyne measurement must be performed within $10^{-5}$ s, so we require at most micro-second precision, which is perfectly achievable. In comparison, we calculated $\bar{k}_{\text{FP}} = 2.30$ for the levitated microsphere, which allows for a very comfortable $\approx 0.19$ s window.

In spite of these challenges, optomechanical systems come with a number of advantages. They can remain stationary while performing the measurement, in contrast to on-chip BECs or BEC fountains which need to be launched, and the short cycle time of optomechanical systems allows for a large number of measurements to be performed very quickly. An additional point which we did not elaborate on above is that the spatial resolution of optomechanical systems will be extremely high since the oscillator is displaced only by a minuscule distance. As a result, it will be possible to determine very fine local variations in $g$, something which is not possible using larger systems. The scheme presented in this work also allows for the creation of macroscopic spatial superpositions, which, as pointed out in[11], is of great interest to testing gravitational collapse models (see for example[53–55]).

Before we conclude, let us now briefly discuss the underlying physical differences between atom interferometry and optomechanical systems for the purpose of gravimetry. We estimate that the QFI for atom interferometry is given by $H_Q(T) \sim n^2 T^4 k_c^2$ up to an unknown geometric factor, where $n$ is the number of photons that deliver a momentum kick to the atoms, $T$ is the total time over which the gravimetric phase is accumulated, and $k_c$ is the laser wavevector. See Supplementary Note 4 for the derivation. If we compare this to the Fisher information for

optomechanical systems, we find that the Fabry–Perot cavity has a QFI that is larger by an enhancement factor $\xi_{\text{FP}} \sim c^2/(nL^2\omega_m^2)$, with $c$ being the speed of light. This is due to the cavity confinement, whereby each photon interacts with the oscillator $c/(2L\omega_m)$ times per oscillation cycle, which is also the time period over which the gravimetric phase is accumulated. For the levitated nanosphere, we find a $\xi_{\text{Lev}} = \xi_{\text{FP}} P^2/(\varepsilon_0 V_c)^2$, where, again, for a micro-object containing $\sim 10^{13}$ atoms, the polarizability $P$ is much higher than that of a single atom. In practice, however, both of the enhancement factors will be damped by a factor $\sim 1/(\omega_m T)^4$ with respect to atom interferometry as the time of atomic interferometry $T$ is typically larger than the time $1/\omega_m$ of our scheme. Thus the sensitivity $\Delta g$ is seen to improve by a factor of $\sqrt{n}L\omega_m^3 T^2/c \sim \sqrt{n} \times 10^{-4}$ in our optomechanical scheme with respect to atomic interferometers. As $n$ increases, the differences level out. However, different initial states in an optomechanical system, such as the superposition of two Fock states, will also scale with $n^2$ (see Supplementary Note 5). Strictly speaking, the enhancement is valid for when the cavity field remains coherent for the time $1/\omega_m$ over which our phase accumulation, i.e., $\kappa \ll \omega_m$ (the resolved side-band regime). However, our numerical results indicate that even in the presence of finite decoherence, say, $\kappa \sim 0.1 \times \omega_m$, the Fisher information is lowered only by a factor of about 10 compared to the case of loss-less cavities. Finally, we can also compare the treatment presented in this work to a position measurement of a classical oscillator that has been displaced due to gravity. While a classical treatment of the problem returns a preliminary measurement sensitivity similar to what we have derived in this work, it fails to take into account effects such as radiation pressure and the full quantum nature of the cavity field. Most importantly, a classical treatment does not utilise the coherent nature of the oscillator, which as we saw above negates any initial thermal noise in the state, and does not allow for the inclusion of other quantum states, such as squeezed states. After completing this work, the authors became aware of similar work carried out by Armata, Latmiral, Plato and Kim [58].

## Methods

**Dimensionless Fisher Information.** For any numerical calculations, we must ensure that the numerical quantities are dimensionless. In order to calculate $I_F(t)$ under decoherence, we separate Eq. 13 above into a dimensionful and dimensionless part by writing

$$I_F(t) = \left(\frac{\partial \bar{g}}{\partial g}\right)^2 \int dx_\lambda \frac{1}{p(x_\lambda|g)} \left(\frac{\partial p(x_\lambda|g)}{\partial \bar{g}}\right)^2. \quad (21)$$

Here, $\partial \bar{g}/\partial g = \cos\theta \sqrt{m/(2\hbar\omega_m^3)}$ is a dimensionful prefactor. The remaining integral

$$\bar{I}_F(t) = \int dx_\lambda \frac{1}{p(x_\lambda|g)} \left(\frac{\partial p(x_\lambda|g)}{\partial \bar{g}}\right)^2, \quad (22)$$

is now dimensionless and is what we will evaluate numerically. A final estimate for $\Delta g$ can then be obtained by multiplying the value for $\bar{I}_F(t)$ by $\cos^2\theta\, m/(2\hbar\omega_m^3)$, but as this is only a rescaling we chose to only present the results for $\bar{I}_F(t)$ for clarity. Note that for the choice of $\bar{k} = \bar{g} = 1$, this dimensional prefactor is equal to $\cos^2\theta\, m/(2\hbar\omega_m^3) = 1/g$.

**Numerical methods.** To evolve the system, we use the Python library Qutip[56] and a 4th order Runge–Kutta–Fehlberg method[57] for verification.

If we wish to compute the CFI for states undergoing decoherence, Eq. 15 is no longer valid and we must evolve the state numerically. We do so by computing the dimensionless part $\bar{I}_F(t)$ in Eq. 20 for a mixed state $\rho(t)$. The probability distribution $p(x|g)$ is easy to obtain numerically, since any operator has a finite matrix representation from which we can obtain the eigenstates and use these as our POVM elements. For example, we can define a position operator $\hat{x}_C$ as a finite-dimensional matrix and solve for its eigenstates.

To obtain $\bar{I}_F(t)$ we must also compute the derivative $\partial p(x|g)/\partial \bar{g}$. This can be done in a number of ways. The simplest one is to use a higher-order method of the central difference theorem. We obtained good and accurate results with the 4th

order five-point method. For a function $f(x)$ with parameter $x$ and step-size $h$, the first derivative with this method is given by

$$f'(x) = [-f(x+2h) + 8f(x+h)$$

$$-8f(x-h) + f(x-2h)]/(12h) + \mathcal{O}(h^4). \quad (23)$$

As this method requires five data point, it is an expensive computation. This was our preferred numerical method as computing the CFI can still be done within reasonable timescales using the optimised master equation solver provided by the Qutip library. It does however contain two different sources of numerical errors: errors in the solver and errors that originate from the cut-off in the numerical derivative.

To verify that the error introduced by the numerical differentiation is not severely affecting the results, we used an additional method which provides an exact result. We reproduce it here for completion and in the hope that it might benefit others. We start by noting that as long as the POVM element $\Pi_x$ does not depend on $\bar{g}$, we can write the derivative as

$$\frac{\partial p(g,x)}{\partial \bar{g}} = \frac{\partial \rho(g)}{\partial \bar{g}} \Pi_x, \quad (24)$$

Note that we are differentiating with respect to $\bar{g}$ instead of $g$ and that we have suppressed the dependence of $t$ for clarity. This statement also holds for subsystems of $\rho(g)$, which we can see by noting that the derivative distributes over a joint separable system $\rho_{AB} = \rho_A \otimes \rho_B$ as

$$\frac{\partial \rho_{AB}}{\partial \bar{g}} = \frac{\partial \rho_A}{\partial \bar{g}} \otimes \rho_B + \rho_A \otimes \frac{\partial \rho_B}{\partial \bar{g}}. \quad (25)$$

Performing a measurement with $\Pi_x$ that only acts on subsystem $A$ then gives

$$\text{tr}_B \left[ \frac{\partial \rho_{AB}}{\partial \bar{g}} \Pi_x \right] = \text{tr}_B \left[ \frac{\partial \rho_A}{\partial \bar{g}} \Pi_x \otimes \rho_B \right] + \text{tr}_B \left[ \rho_A \Pi_x \otimes \frac{\partial \rho_B}{\partial \bar{g}} \right]. \quad (26)$$

The second term reduces to zero because $\text{tr}[\partial_{\bar{g}} \rho_B] = \partial_{\bar{g}} \text{tr}[\rho_B] = 0$. While we have shown this for separable states, the same argument can be extended to entangled states by linearity.

In order to obtain the evolution for this state, we must now solve a modified version of the master equation. That is, given the Lindblad equation in Eq. 19,

$$\dot{\rho}(\bar{g}) = -\frac{i}{\hbar}\left[\hat{H}(\bar{g}), \rho(\bar{g})\right] + \hat{L}\rho(\bar{g})\hat{L}^\dagger - \frac{1}{2}\{\rho(\bar{g}), \hat{L}^\dagger \hat{L}\}, \quad (27)$$

where $\{\cdot, \cdot\}$ denotes the anti-commutator, we now differentiate both sides with respect to $\bar{g}$ to obtain

$$\partial_{\bar{g}} \dot{\rho}(\bar{g}) = -\frac{i}{\hbar}\left[\partial_{\bar{g}}\hat{H}(\bar{g}), \rho(\bar{g})\right] - \frac{i}{\hbar}\left[\hat{H}(\bar{g}), \partial_{\bar{g}}\rho(\bar{g})\right]$$

$$-\frac{1}{2}\{\partial_{\bar{g}}\rho(\bar{g}), \hat{L}^\dagger\hat{L}\}, \quad (28)$$

where we have again used the notation $\partial_{\bar{g}} = \partial/\partial\bar{g}$. A more complicated form is obtained if the Lindblad operators $\hat{L}$ depend on $\bar{g}$, which here is not the case. In coupled form, we can write

$$\frac{d}{dt}\begin{pmatrix} \rho \\ \partial_{\bar{g}}\rho \end{pmatrix} = \begin{pmatrix} -\frac{i}{\hbar}[\hat{H}, \rho] + (\hat{L}\rho\hat{L}^\dagger - \frac{1}{2}\{\rho, \hat{L}^\dagger\hat{L}\}) \\ -\frac{i}{\hbar}\left([\partial_{\bar{g}}\hat{H}, \rho] + [\hat{H}, \partial_{\bar{g}}\rho]\right) + \hat{L}\partial_{\bar{g}}\rho\hat{L}^\dagger - \frac{1}{2}\{\partial_{\bar{g}}, \rho\hat{L}^\dagger\hat{L}\} \end{pmatrix}. \quad (29)$$

The system can be solved using any standard higher-order method, such as the family of Runge–Kutta ODE solvers. Note that the Qutip Master Equation solver cannot be used as Eq. 27 is not in standard Hamiltonian form.

Once the time-evolved state $\partial\rho/\partial\bar{g}$ has been obtained, we proceed as usual to compute the probability distribution with the set of POVM elements $\{\Pi_x\}$. With this method, we avoid round-off errors that appear in the five-point numerical derivative above.

**Numerical stability**. Let us make a few remarks regarding the numerical stability of the simulation. We start by considering the nature of coherent states and how they are represented numerically. Coherent states have support on infinite Hilbert spaces, whereas numerically we must work with finite matrices. It is therefore necessary to introduce a cut-off in the dimension used to represent the state. This leads to a gradual loss of coherence as information is pushed beyond the cut-off. In other words, numerically we use a finite Hilbert space $\mathcal{H}$, meaning that we truncate

the space by letting $\hat{a}^\dagger|N-1\rangle = 0$, where $\dim(\mathcal{H}) = N$. Furthermore, the appearance of $(\hat{a}^\dagger\hat{a})^2$ in $\hat{U}(t)$ causes the system to become anharmonic and numerical instabilities grow fast for Hilbert spaces with small dimension $N < 50$.

The amount of information lost when using smaller Hilbert spaces is difficult to assess, since any good ODE solver will preserve the purity of the state throughout the simulation. Rather, the loss of information can be noted as a gradual deterioration of the trajectory in phase space, with the effect that states fail to return to their original position in phase space at $t = 2\pi$. That is, we require that $\langle\hat{x}(0)\rangle \approx \langle\hat{x}(2\pi)\rangle$ and $\langle\hat{p}(0)\rangle \approx \langle\hat{p}(2\pi)\rangle$ for the simulation to be deemed stable. This is however only true for noiseless evolution.

The system dynamics depend strongly on the dimensionless constants $\bar{k}$ and $\bar{g}$. Larger $\bar{k}$ and $\bar{g}$ will cause the system to evolve more rapidly, as evident from their appearance in the phase of the state in Eq. 7. This in turn causes the numerical inaccuracies to accumulate more rapidly. When computing the CFI for mixed states, we restrict our investigations to the parameter range $\bar{k} = \bar{g} = 1$ for precisely this reason.

Finally, it should be noted that we have not provided a full error estimate for any of the results computed here. However, since we are only interested in the general behaviour of the CFI, any small inaccuracies in the numerical estimates will not matter for the results obtained in this work.

**Data availability**. The datasets generated during and/or analysed during the current study are available in the Coherent-states-Fisher-information repository on Github at (https://github.com/sqvarfort/Coherent-states-Fisher-information).

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

## Acknowledgements

The authors would like to thank Abolfazl Bayat, Nathanaël Bullier, Victor Montenegro, Dennis Schlippert, Stephen Stopyra and Doug Plato for useful comments and discussions. This work was supported by the EPSRC Centre for Doctoral Training in Delivering Quantum Technologies and the EPSRC grant EP/N031105/1.

## Author contributions

S.B. initiated the main idea and S.Q. performed the calculations and the numerical simulations. A.S. suggested the model with leaking photons and P.F.B. provided valuable insight into the experimental settings. All authors contributed to the interpretation of data, discussions of experimental implementations and writing of the manuscript.

## Additional information

**Competing interests:** The authors declare no competing interests.

