## [Peer Review File · Nature Communications]

Reviewers' comments:

Reviewer #1 (Remarks to the Author):

The authors present a protocol for gravimetry based on the use of a mechanical oscillator interacting with an optical cavity field.

The gravitational potential adds a linear term to the Hamiltonian and the uncertainty in the determination of the coupling parameter is studied using the classical and quantum Fisher information. Interestingly, the measurement scheme proposed in a routine homodyne detection of the cavity state which saturates the bound, being the optimal measurement.

It is hard for me to assess the quality of the comparison with other experimental protocols, but on the theory side the work constitutes an important contribution to precision sensing.

In addition, the article is written with great care and it is a joy to read.

The presentation is slightly confusing when stating

How is it possible that " So while initial cooling of the oscillator is not required, cooling the system during the evolution will be necessary." as the authors assume predominantly unitary dynamics.

This seems inconsistent. This also seem to be in contradiction with the purity of $\psi(t)$. Only in p.6 the authors provide a estimate and a clarification.

A more rigorous approach would be to resort to dissipative quantum metrology [Phys. Rev. Lett. 112, 120405 (2014); hys. Rev. Lett. 119, 010403 (2017)] which can pave the way to improvements on the current analysis.

But for this remark, that the authors are encouraged to take into account, the manuscript seems well suited for publication in nature Communications, that I recommend.

Reviewer #2 (Remarks to the Author):

This manuscript investigates theoretically the sensitivity of measurements of gravitational acceleration using cavity optomechanics. g is determined by homodyne detection of the field exiting the cavity in which it interacts dispersively with a movable reflector. Based on a quantum treatment of this canonical interaction the authors compute the classical and quantum Fisher information and show that a homodyne detection measurement scheme can optimize the sensitivity. This ideal sensitivity is computed in the case of a linear Fabry-Perot resonator and for a levitated microsphere and predicted to surpass that currently achieved in atom-interferometry experiments.

The scheme proposed by the authors is interesting and amenable to a number of optomechanical systems. The theoretical derivations appear sound and corroborate similar results obtained in another recent work (ref. 45) published in Phys. Rev. A. The paper is in general clearly written and structured. However, while this proposal addresses a specific theoretical issue which is interesting for the optomechanics community, the results are not so groundbreaking as to warrant publication in Nature Communications. I also find the paramaters chosen in the physical implementations questionable and the ideal sensitivity estimates rather optimistic. I would thus recommend this

work for publication in a more specialized journal.

Specific comments:

- the values of β and α used in fig. 1b should be given for completeness.
- the authors should more quantitatively (numerical simulations) argue why starting with the mechanics in a coherent states and calculating a pure state evolution can be used to investigate the physically realistic case of a high-temperature thermal state oscillator. I assume that they refer to the same kind of dynamics as investigated in ref. 31, but this should be made clearer.
- the discussion of the effects of decoherence studied in the case $\bar{g}=1$ are not convincing, as any realistic cavity would have a $\bar{g} \gg 1$.
- in page 7, 1st column, the authors look at photon decay rates $k \sim 0.05\omega_0$, which for the linear Fabry Perot cavity parameters they choose would result in $k \sim 50/s$, i.e. unrealistically low cavity losses at the 10^{-11} level!

Starting with the comments by reviewer #1:

1. The reviewer stated: *‘The presentation is slightly confusing when stating: “So while initial cooling of the oscillator is not required, cooling the system during the evolution will be necessary.” as the authors assume predominantly unitary dynamics. This seems inconsistent. This also seem to be in contradiction with the purity of $\psi(t)$. Only in p.6 the authors provide a estimate and a clarification. We have removed the comment that caused the confusion and added an explanation of this point when the issue is first raised on p. 3 in the manuscript. We have also clarified further that we do not consider damping to the oscillator in this paper. We have also added an analytical proof in the appendix showing that the final Fisher information is unaffected by the initial thermal state of the oscillator.*
2. The reviewer stated: *‘A more rigorous approach would be to resort to dissipative quantum metrology [Phys. Rev. Lett. 112, 120405 (2014);hys. Rev. Lett. 119, 010403 (2017)] which can pave the way to improvements on the current analysis.’* We have studied both papers but concluded that while especially the first reference is related, the estimation does not enter linearly into the noise operator in our situation, and so is a slightly different scenario. We have instead used our numerical results as guidance to estimate what we believe would be a reasonable estimate for the Fisher information and shown that there are experiments that meet all the requirements for this scheme (see p. 7 in the manuscript). We have cited both papers in the manuscript.

Similarly we have addressed the comments by reviewer #2 as follows:

1. The reviewer stated: *‘However, while this proposal addresses a specific theoretical issue which is interesting for the optomechanics community, the results are not so groundbreaking as to warrant publication in Nature Communications.’* To show that our work is relevant for a broader audience, we have added a calculation for cold atoms trapped in cavities, as well as additional references regarding the use of gravimetry to emphasize that it is a topic of wide interest, including a recent study on the gravity harmonics of Jupiter, hence quantum avenues for this purpose are relevant to a wide scientific and industrial community.
2. The reviewer stated: *‘I also find the paramaters chosen in the physical implementations questionable and the ideal sensitivity estimates rather optimistic.’* All parameters, by which we mean the oscillation frequencies, masses, lengths, etc. that we have chosen for our examples have been experimentally implemented in various optomechanical systems. Numbers for the cold atom case come straight from the paper cited. The ideal sensitivities follow directly from this. We limited our estimates to order-of-magnitude estimates in order to emphasize that they are indeed a rather optimistic proof-of-concept rather than predictions for the final measurement sensitivity. However, we have made efforts to further address this statement, as presenting an experimentally viable scheme is of obvious interest to the community. As such, we have added another full section starting on p. 7. which discusses the conditions under which the scheme performs well. We have also suggested and reference methods by which decoherence can be further reduced for these systems. Naturally, additional losses are expected compared to the one we consider, and so the realistic sensitivity will be lower, but as stated before the main purpose of this paper is a proof-of-concept showing that optomechanical systems can be at least as good as more established atom interferometry schemes.
3. The reviewer stated: *‘The values of beta and alpha used in fig. 1b should be given for completeness.’* This has been corrected.

4. The reviewer stated: *‘The authors should more quantitatively (numerical simulations) argue why starting with the mechanics in a coherent states and calculating a pure state evolution can be used to investigate the physically realistic case of a high-temperature thermal state oscillator. I assume that they refer to the same kind of dynamics as investigated in ref. 31, but this should be made clearer.* To fully address this point we have added an analytical proof in the appendix which shows that the Fisher information remains the same even when we start with a thermal mixture of coherent states, and thus we do not require cooling to the ground state. We refer to this proof at several points throughout the paper for clarity.
5. The reviewer stated: *‘The discussion of the effects of decoherence studied in the case $bag=1$ are not convincing, as any realistic cavity would have a $\bar{g} \gg 1$.’* Indeed, realistic parameters would yield $\bar{g} \sim 10^7$, but this is essentially impossible to simulate computationally if one wishes to take into account the full non-linear dynamics *as well as* decoherence effects. For pure states, we can computationally manage about $\bar{g} = 10$ before the system becomes numerically unstable. This is essentially due to the truncation of the Hilbert space and is limited by computer memory. As such, we cannot address this point numerically. However, we have added another computation with $\bar{\kappa} = 0.1$ and shown that there are systems which would allow us to obtain these parameters, see the next point.
6. The reviewer stated: *‘In page 7, 1st column, the authors look at photon decay rates $k \sim 0.05\omega_O$, which for the linear Fabry Perot cavity parameters they choose would result in $k \sim 50/s$, i.e. unrealistically low cavity losses at the 10^{-11} level!’* We are unsure of what the last number refers to, but we have attempted to address this in a similar fashion to the point above. We have shown that $\bar{\kappa} = 0.1$ can be achieved by systems with $\kappa = 660$ Hz and $\omega_m = 6600$ Hz, corresponding a system which has already been experimentally implemented. This can be found on p. 7 in the manuscript As such, the losses are not unrealistic and the scheme should be viable. We also calculate the ideal sensitivity for this setup and find $\Delta g = 6.80 \times 10^{-8}$ ms⁻², which is similar to the other experimental systems considered.

In addition to these points, we have made the following changes: Clarified notation in various places, improved the text, added additional analysis on the behaviour of classical Fisher information, improved the plots and added additional references.

Reviewers' comments:

Reviewer #1 (Remarks to the Author):

The authors have addressed my previous comments and suggestions thoroughly improving the readability of the work. The work constitutes an important contribution to precision sensing and gravimetry using optomechanics and I recommend the publication of the manuscript in its current form.

Reviewer #2 (Remarks to the Author):

After reading the revised manuscript and the authors' response to the referees' comments I am still of the opinion that the results are not groundbreaking enough so as to warrant publication in Nature Communications (similar results obtained in ref. 57 using a similar methodology were published in PRA).

While the discussion of the physical systems, including the addition of the cold atom implementation, has improved, I still find it unconvincing. For instance, in page 7, 2nd column, the authors use a 10 micron-long Fabry-Perot resonator in their example of a microgram, kHz oscillating mirror to calculate a sensitivity at the 10^{-15} m/s² level. However, later, when discussing how to achieve the necessary resolved sideband condition, they use a 10cm long cavity with a state-of-the-art finesse $>10^6$, yielding a much lower sensitivity at the 10^{-8} m/s² level, even when assuming a much lighter mirror than used in the cited work. All in all, the numbers presented are not convincing from an experimental point of view, and frankly misleading in places.

Further to the comments provided to us by one of the reviewers, we would like to submit the following improved manuscript. We have made efforts to address the concerns expressed by reviewer #2 as follows:

1. The reviewer stated: *'After reading the revised manuscript and the authors' response to the referees' comments I am still of the opinion that the results are not groundbreaking enough so as to warrant publication in Nature Communications (similar results obtained in ref. 57 using a similar methodology were published in PRA).'*: We have made no additional changes to address this concern, but wish to draw attention to the fact that we have obtained additional results which were not presented in the referenced work. These results include: an analytical proof of the optimality of the homodyne detection scheme, additional analysis of the classical Fisher information (which yields the time-scale over which the measurement must be performed), simulations of noisy systems, a photon-leakage toy-model and substantial discussions of an experimental implementation of the system.
2. The reviewer stated: *'While the discussion of the physical systems, including the addition of the cold atom implementation, has improved, I still find it unconvincing. For instance, in page 7, 2nd column, the authors use a 10 micron-long Fabry-Perot resonator in their example of a microgram, kHz oscillating mirror to calculate a sensitivity at the 10^{-15} m/s² level. However, later, when discussing how to achieve the necessary resolved sideband condition, they use a 10cm long cavity with a state-of-the-art finesse $> 10^6$, yielding a much lower sensitivity at the 10^{-8} m/s² level, even when assuming a much lighter mirror than used in the cited work. All in all, the numbers presented are not convincing from an experimental point of view, and frankly misleading in places.'* Firstly, we would like to draw attention to the fact that the QFI (Quantum Fisher Information) is independent of the mass of the mirror, thus the assumption of a smaller mass does not change the measurement sensitivity. To further address this concern, we have added several clarifying statements to the manuscript on p.7, first column, and p.8, first and second column. In summary, we state that the initial calculations of the ideal QFI and thus fundamental sensitivity Δg is meant to showcase the potential of optomechanical systems, for which we have used state-of-the-art parameters. However, since the low decoherence rates required for this scheme have not yet been implemented for the parameters we chose, we calculate the QFI for an already realised system to show what kind of measurement sensitivities are already in range today. As technology improves, we expect that the parameter regime that we considered for the ideal cases will become increasingly accessible. We have made slight changes to the wording in the abstract to be completely clear about this point. We hope that these additional statements will clarify the motivation behind each calculation.